# The Role of HSP90 Inhibitors in the Treatment of Cardiovascular Diseases

**DOI:** 10.3390/cells11213444

**Published:** 2022-10-31

**Authors:** Shiyu Qi, Guang Yi, Kun Yu, Chong Feng, Shoulong Deng

**Affiliations:** 1College of Animal Science and Technology, China Agricultural University, Beijing 100193, China; 2Aurora Bioscience Co., Ltd., Suzhou 215000, China; 3National Health Commission (NHC) of China Key Laboratory of Human Disease Comparative Medicine, Institute of Laboratory Animal Sciences, Chinese Academy of Medical Sciences and Comparative Medicine Center, Peking Union Medical College, Beijing 100021, China

**Keywords:** cardiovascular disease, heart failure, heat shock proteins, HSP90 inhibitors, therapeutic

## Abstract

Cardiovascular disease is the result of complicated pathophysiological processes in the tissues that make up the blood vessels and heart. Heat shock protein 90 (HSP90) can interact with 10% of the proteome and is the most widely studied molecular chaperone in recent years. HSP90 is extensively involved in the regulation of protein folding and intracellular protein stability, making HSP90 a hopeful target for the treatment of multiple cardiovascular diseases. Numerous client proteins of HSP90 have been identified in known cardiac disease pathways, including MAPK signaling, PI3K/AKT (PKB)/mTOR, and TNF-α signaling. Therefore, these pathways can be controlled by regulating HSP90. Among them, the activity of HSP90 can be regulated via numerous inhibitors. In this review, first, we will discuss the function of HSP90 and its role in pathological pathways. In addition, HSP90 plays a significant role in most cardiovascular diseases, including hypertension, pulmonary venous hypertension, atherosclerosis, and heart failure; next we will focus on this part. Finally, we will summarize the currently known HSP90 inhibitors and their potential in the treatment of heart disease.

## 1. Introduction

In 1962, Italian scientist Ritossa and colleagues discovered that exposing fruit fly larval saliva to high temperatures produced a characteristic swelling of the chromosomes [1]. Ritossa believed that the expansion of chromosomes was the result of the activation of corresponding genes in response to heat stress leading to the increased expression of specific proteins, which are now known to encode proteins called heat shock proteins (HSPs) [2]. HSPs is a group of highly conserved proteins; both the simplest prokaryotes and complex mammals contain HSPs [3]. Unlike other mostly cellular proteins, HSP expression and synthesis are increased following stresses such as oxidative stress, nutrient deficiency, UV radiation, chemicals, viruses, and ischemia-reperfusion injury [4,5,6,7]. HSPs are ubiquitously expressed in normally unstressed cells [8]. Expression is low under normal physiological conditions, accounting for 5–10% of the total protein content under healthy growth conditions [9,10], increasing up to 15% under stress conditions [5].

As molecular chaperones, their synthesis is related to protein misfolding and denaturation. HSPs can guide the synthesis of new polypeptides through protein folding, avoid protein misfolding and aggregation, and can restore the three-dimensional protein structure after partial deformation to achieve normal physiological functioning [4,11]. HSPs are also involved in controlling the transport of certain regulatory proteins and intracellular proteins. HSPs are divided into different families according to their molecular weight: namely, 110, 90, 70, 60, 40 kDa and low molecular weight families. Among them, the HSP90 molecular chaperone is the most abundant in the cytoplasm and plays a leading role in the homeostasis of cells [12]. The conformational flexibility of HSP90 and its various co-chaperone complexes contribute to its functional diversity and enable HSP90 to assist a wide range of substrates [13]. HSP90 contributes to the maturation of a group of substrate proteins called clients. Client proteins include tyrosine kinases (e.g., Akt and MEK), hormone receptors, structural proteins (tubulin and actin), hypoxia-inducible factor 1α (HIF-1α), HSF1, and HSP70 [14,15]. HSP90 can be thought of as a “molecular buffer” that maintains the general balance and fidelity of cell signaling, and HSP90 inhibitors are being investigated for the treatment of various conditions [16]. In addition to their development in cancer therapy, HSP90 inhibitors have important roles in the treatment of many inflammatory diseases [6,7,17,18,19,20]. At the same time, they can be used as a target for the treatment of cardiovascular disease (CVD) [21,22,23,24].

Data from the World Health Organization show that cardiovascular disease ranks first in morbidity and mortality worldwide. Deaths from CVD in 2019 were 17.8 million, ten years from now the death toll is expected to rise to 23 million [25]. CVDs are the collective term for diseases of the heart and blood vessels, including hypertension, pulmonary hypertension, atherosclerosis, cardiac fibrosis, hypertrophy, heart failure, etc. [26]. Mortality from cardiovascular disease is largely determined by disease caused by atherosclerosis. It is an autoimmune chronic disease and a chronic inflammatory disease [27]. Other key causes of cardiovascular disease are related to oxidative stress, lipid metabolism, metabolic and molecular changes in inflammation, and myocardial dysfunction [28].

The cardiovascular system, consisting of the heart and blood vessels, plays a central role in human physiology, and its main contribution is the continuous supply of nutrients and blood to the remaining tissues to support their metabolic activities [29]. Therefore, cardiac cells have very strict requirements for metabolites, and in the organelle protein turnover, synthesis, and quality are strictly controlled [30]. Increased reactive oxygen species, oxidation, and mechanical stress are important features of CVD, resulting in the misfolding or damage of proteins and the accumulation of DNA mutations in mitochondria [31]. HSP90 is involved in the stabilization of proteins against heat stress and contributes to protein degradation; after interacting with misfolded proteins, it can reduce their aggregation with other misfolded proteins, further reducing the aggregation of misfolded proteins by polyubiquitination and the 26S proteasome. It is associated with protein degradation and destruction [2,32]. Therefore, HSP90 plays an important role in the treatment and research of heart and vascular diseases.

In this review, we summarize the relevant knowledge of HSP90 in several major CVDs, such as hypertension, pulmonary venous hypertension, atherosclerosis, heart failure, and many other cardiovascular diseases. The function of HSP90 and its interactions in CVDs are discussed, which will provide insight into the approaches that support HSP90 in the treatment of cardiovascular disease. 

## 2. Subtypes and Molecular Properties of HSP90 and Its Secretory Pathway

### 2.1. Structure of HSP90

The HSP90 family of molecular chaperones is the most abundant in the cytoplasm. In mammals, the four known isoforms are: HSP90α, HSP90β, ER-localized glyco-responsive protein 94 (Grp94), and mitochondrial TNF receptor-related protein 1 or HSP90N (TRAP1) [33]. The functional domain of HSP90 is mainly divided into three parts (N-terminal domain, C-terminal domain, and intermediate domain) (Figure 1). These three distinct domains may provide stability for HSP90s interactions with misfolded proteins produced by heat stress and inflammation, reducing their aggregation with other misfolded proteins [8,12,34,35,36]. The three functional domains are connected by a linker that enables folding and helps HSP90 to reach the active conformation [37]. Linking of the N-terminal and intermediate domains in eukaryotes is achieved through a charged linker domain [38]. This structure is critical in the chaperone function, interaction, and flexibility [38]. The charged linker domains vary in length and amino acid sequence composition [39]. The three functional domains in HSP90 have different divisions of labor. The N-terminal domain is called the nucleotide binding site and binds to ATP. The C-terminal domain is responsible for protein dimerization and contains special motifs MEEVD or KDEL, depending on the HSP90 isoform and its cellular location either in the cytoplasm or the ER [2].

#### 2.1.1. HSP90α and HSP90β

HSP90α and HSP90β function in the cytoplasmic matrix and may originate from duplicated genes [34,40]. Their sequences are highly homologous, with 85% homology. There are differences in specific protein sequences, so the functions of the two isoforms are also different [41]. Despite being subtypes, HSP90α and HSP90β have some changes in specific parts of their protein sequences. In addition, the length of HSP90α (732) is different from that of HSPβ (724). The α and β isoforms vary in specific regions along the protein sequence, suggesting diverse functions of the two isoforms [2]. Due to the similarity between the two isoforms, most client proteins are able to bind to both of them.

#### 2.1.2. GRP94

GRP94 is another isoform of HSP90 and shares 50% homology with cytoplasmic HSP90 [40]. It is most abundant in the endoplasmic reticulum, and it is also known as endoplasmic [42,43]. GRP94 shares some similarities with its cytoplasmic homolog, including binding and hydrolyzing ATP, but it has specific protein clients. GRP94 is different from HSP90 in the cytoplasm [44]. GRP94 exists in extended, less extended, and closed conformations that facilitate nucleotide binding to client proteins [45,46,47]. GRP4 also contains three different functional domains, the N-terminal domain, C-terminal domain, and intermediate domain, but its length is shorter than that of HSP90 in the cytoplasm, and the terminal has a KDEL retention signal [48]. Compared with cytoplasmic HSP90, GRP94 has a shorter charged linker region, is rich in lysine residues and calcium-binding sites, and exhibits stronger acidity [35]. Furthermore, the charged linker domain regulates conformational changes upon ATP hydrolysis, and its absence greatly affects GRP94 function and interaction with clients [49]. GRP4 contains eleven low-affinity and four high-affinity calcium binding sites, which can regulate the activity and conformation of GRP4 through the degree of calcium binding, and further regulate the level of GRP4 and client peptides [50].

#### 2.1.3. TRAP1

TRAP1, another isoform of HSP90, is mainly found in the mitochondrial matrix [50,51,52,53]. It has 50% homology with cytoplasmic HSP90, and the composition of its functional domains is similar. It also consists of three domains: NTD, MD, and CTD [1,40]. Moreover, its NTD can also bind ATP [54], and the binding affinity of TRAP1 to ATP is 10 times higher than that of the other HSP90 proteins. Additionally, heat shock can increase TRAP1 expression 200-fold [55,56]. TRAP1 has a mitochondrial targeting sequence of 59 amino acids which is cleaved off upon introduction into mitochondria [54]. On the other hand, compared with the protein HSP90, TRAP1 lacks the C-terminal MEEVD motif and the domain connecting the C-terminal domain: the intermediate domain [57]. Furthermore, certain co-partners, such as p23 and Hop, etc., are not required for their functioning [58].

### 2.2. Potential Pathways for HSP90 to Enter the Bloodstream

#### 2.2.1. Secretion of Viable Cardiomyocytes

HSP90α, a subtype of the HSP90 family, can be secreted extracellularly and has been implicated in physiological and pathological processes such as wound healing, cancer, infectious diseases, and diabetes. Studies have shown that HSP90α can be secreted by endothelial cells upon stimulation by angiogenic cytokines, chemokines, and extracellular matrix (ECM) proteins in in vitro dermal cells and in in vivo wound healing models as well as in mouse skin wound healing models. Secreted HSP90α localizes to the leading edge of migrating endothelial cells and promotes angiogenesis in wound healing [59].

#### 2.2.2. HSP90 as a Component of Exosomes Secreted by Cardiac Cells into Blood and HSP90 Being Released by Dying Cardiac Cells

The study by Hunter-Lavin et al. showed that the release of HSP70 from peripheral blood mononuclear cells into blood or culture medium is not due to cellular damage, but via a non-canonical pathway involving lysosomal lipid rafts [60]. Most HSPs do not contain a consensus peptide signal and, thus, cannot secrete proteins via the canonical ER–Golgi pathway. Furthermore, inhibition of the ER–Golgi pathway by typical inhibitors such as brefeldin A does not inhibit live rat embryo cells to release HSP70 [61]. These observations suggest that HSPs may be actively exported through non-classical or unconventional secretory pathways in living cells [62]. Exosomes are nanoscale extracellular vesicles secreted by various cell types that contain proteins, nucleic acids, or other cargoes which play an important role in communication between different cells. As an important type of unconventional protein secretion (UPS), exosomes are also involved in the secretion of extracellular HSPs [63]. Therefore, it is likely that HSP90 is also secreted into the blood through the UPS.

#### 2.2.3. Necrotic Cells Can Passively Release HSPs into the Extracellular Environment

Tytell et al. first provided evidence that HSPs can be released and taken up in 1986. They reported that HSPs can be transferred from the glia to the axon [64]. Subsequent studies have shown that HSPs are selectively released through non-canonical secretion mechanisms, but have been ignored until 2000 when Srivastava et al. found that necrotic but not apoptotic cells can release HSPs (such as gp96, calreticulin, HSP90, and HSP70) into the extracellular environment, thereby stimulating antigen-presenting cells (APCs) such as macrophages and dendritic cells which secrete several different cytokines through the nuclear factor kappa B (NF-κB) pathway [65]. Therefore, necrotic cells can passively release HSPs into the extracellular environment, and these extracellular HSPs can stimulate monocyte activation through the NF-κB pathway to produce cytokines, thereby regulating the immune system [19].

## 3. HSP90 and Cardiovascular Disease

### 3.1. HSP90 and Hypertension

Hypertension is one of the primary causes of death in the world, and it is an important contributor and risk factor for cardiovascular morbidity and mortality. In addition, it can lead to many complications [66].

Hypertension-induced endothelial dysfunction is associated with impaired nitric oxide (NO) bioavailability regulated through the interaction between nitric oxide synthase and HSPs [67]. It is well known that nitric oxide is one of the key factors in maintaining normal blood pressure. Arterial hypertension is associated with impaired NO function [68,69]. One of the key client proteins of HSP90 is endothelial NOS. HSP90 binds to eNOS, thereby determining its correct folding and function [70]. In an animal model of spontaneous hypertension, HSP90 expression was higher in the left ventricle of tonic rats compared with normotensive control animals. Interestingly, all isoforms of NOS (neuronal (nNOS), inducible (iNOS) and eNOS) were initially down-regulated but were eventually significantly increased in the left ventricle of spontaneously hypertensive rats. Increased expression of HSP90α compensates for impaired eNOS function associated with reduced nitric oxide bioavailability [67]. Increased HSP90α concentrations in patients with arterial hypertension may be a compensatory mechanism for impaired nitric oxide bioavailability [67], confirming that HSP90α can serve as an early marker of hypertension-related endothelial injury [67]. In addition, it was also found that this study mainly focused on HSP90 in serum. According to previous studies in tumors, it was found that when a tumor occurs HSP90α can be specifically secreted by tumor cells into the blood outside the cell, and its concentration in plasma is related to malignant tumors. The occurrence and progress are closely related [71]. However, the secretory pathway or mechanism in cardiovascular disease needs further study. Studies have shown that HSP90 mediates AngII-induced vascular smooth muscle cell (VSMC) proliferation and remodeling, and brain microvascular injury during hypertension [20,72,73]. Adventitial remodeling is involved in the evolution of several vascular diseases including hypertension [74]. This remodeling is manifested by exogenous thickening, increased numbers of fibroblasts (the main cellular component of exogenous cells), and a phenotypic change from exogenous fibroblasts (AFs) to myofibroblasts (MFs) [75]. Among them, the phenotypic transformation of AF is crucial in vascular remodeling. An HSP90-regulated pathway involving calcineurin (CN) and Drp1 mediates angiotensin II (AngII)-induced AF phenotype switching and adventitial remodeling [76]. Inhibition of HSP90 improves CN expression and subsequent Drp1 activation, ultimately attenuating the adventitial remodeling of AngII-induced hypertension, suggesting that HSP90 inhibition is a novel therapeutic approach for the remodeling of hypertension [76]. The results of an animal model of AngII-induced hypertension demonstrate that inhibition of HSP90 effectively attenuates AngII-induced phenotype switching in atrial fibrillation and subsequent migration and proliferation [76]. A study compared mice with different diet-induced hypertension regimens, with the addition of repetitive hyperthermia in some groups [77]. The results of the study found that mice treated with a high-salt diet developed cardiac hypertrophy and fibrosis, whereas repetitive hyperthermia attenuated salt-induced cardiac hypertrophy, myocardial and perivascular fibrosis, and blood pressure elevation. In addition, the levels of HSP60, HSP70, and HSP90 were all increased in mice subjected to repeated hyperthermia [77].

### 3.2. HSP90 and PAH

Pulmonary arterial hypertension (PAH) is high blood pressure in the arteries that flow from the heart to the lungs and is a multifactorial disease [66]. Unlike systemic hypertension, PAH is primarily caused by vascular remodeling and vasoconstriction. Progressive stenosis of the pulmonary arteries and reduced compliance of the pulmonary arterial system are the central aspects of PAH. Furthermore, right ventricular (RV) overload and RV failure finally result due to elevated pulmonary vascular resistance and pulmonary artery pressure [78]. In this regard, the HSP90 chaperone mechanism has become a hopeful axis [66].

Recently, several reports have implicated HSP90 in vascular remodeling, which has a major impact on PAH. However, the exact pathogenesis of PAH remains to be elucidated [74,79,80]. A study has shown that the levels of HSP90 are increased in both the plasma membrane wall and the membrane wall of pulmonary arterioles in PAH patients [74]. HSP90 is involved in regulating the functions of many proteins involved in PAH development, such as AMPK and sGC [81,82]. An excessive spread of pulmonary artery smooth muscle cells (PASMCs) in the lining of the pulmonary artery may be one of the most prominent features of PAH [83,84]. Studies have found that HSP90 inhibitors can improve pulmonary arteriole remodeling by inhibiting the extensive proliferation of PASMCs; thus, inhibiting HSP90 may be an effective treatment for PAH [74]. There are clear similarities between PAH and cancer [85]. HSP90 is highly expressed not only in the cytoplasmic matrix, but also in subcellular pools, such as in the mitochondria of tumor cells where it promotes cell survival, but is absent in normal cells [86,87]. Similar to cancer cells, PASMCs from PAH patients contain significant amounts of mitochondrial HSP90 (mtHSP90) [88]. It has been found that HSP90 preferentially accumulates in the mitochondria of PAH-PASMCs in response to stress, and the accumulation of mtHSP90 is a feature of PAH-PASMCs and an essential regulator of mitochondrial homeostasis, contributing to PAH vascular remodeling [88]. Caveolin-1 (cav-1) may play a significant role in the pathogenesis of PAH. Some studies have shown that endothelial cav-1 is significantly lost in human PAH [89,90]. eNOS and cav-1 form a closed complex in vitro and in vivo, leading to their dysfunction during hypoxia [91,92]. In addition, the binding of eNOS to HSP90 is also disrupted under hypoxia conditions [93]. Therefore, hypoxia-induced cav-1/eNOS complex formation may impair the binding of HSP90 and eNOS partly, resulting in injured vascular relaxation [94]. A study in a monocrotaline-induced rat model of PH found that the use of HSP90 inhibitors alleviated the progression of PAH, manifested by decreased pulmonary arterial pressure and loss of right ventricular hypertrophy [74].

### 3.3. HSP90 and Atherosclerosis

Atherosclerosis is a type of cardiovascular disease that occurs in the intimal layer of large- and medium-sized arteries caused by lipid deposition, especially low-density lipoprotein (LDL) associated with immune cell infiltration and arterial wall remodeling [95,96]. It is considered an autoimmune chronic disease that usually culminates due to the formation of plaques [25]. The plaque will cause the diameter of the blood vessels to shrink, depriving the heart muscle of sufficient oxygen and sufficient blood flow, resulting in myocardial ischemia [25]. Additionally, if the plaque ruptures suddenly, coronary artery occlusion may occur leading to myocardial infarction [25].

The study found that HSP90 levels were increased in the serum of patients with carotid atherosclerosis, and that HSP90 was overexpressed in human atherosclerotic plaques [97,98]. LDL receptor-related protein 1 (LRP1) is the first identified as an HSP90 receptor [99]. LRP1 can bind and internalize various ligands, for example, activated α2-macroglobulin and apoE, which is an endocytic receptor [99]. It can regulate the migration and proliferation of a variety of cells including macrophages, regulate intracellular signaling, and play a key role in the formation of atherosclerotic plaques [99]. The phagocytic scavenger receptor is a trimeric membrane protein that binds LDL, and HSP90 has been shown to bind to its cytoplasmic domain portion [100]. Therefore, it is speculated that HSP90 may exert its effect on atherosclerosis by affecting LDL metabolism [101].

HSP90 expression is associated with features of plaque instability in advanced human lesions. HSP90 inhibitors reduce inflammatory responses in atherosclerosis [102]. In cultured vascular smooth muscle cells and human macrophages, STAT and NF-κB activation and chemokine expression were reduced with HSP90 inhibitors [102]. In hyperlipidemic apoE deficient mice, treatments with HSP90 inhibitors also showed decreased NF-κB and STAT activation, decreased lipid and macrophage content, and decreased atherosclerotic plaque lesions [102]. In addition, HSP90 expression is also associated with features of plaque instability in advanced lesions [102]. HSP90 is not only associated with NF-κB, but also up-regulates plaque matrix metalloproteinase 8 (MMP-8), which plays a role in the degradation of collagen types I, II, and III, and therefore may have an effect on changes in the plaque vulnerability index [17]. In addition, the study also found that the beneficial effect of inhibiting HSP90 activation was related to the overexpression of HSP70 [95]. In summary, these results indicate that HSP90 may have an effect on promoting atherosclerosis.

### 3.4. HSP90 and Heart Failure

Heart failure (HF) is the main cause of death from cardiovascular disease. HF is not an independent disease, but a common terminal pathway of various heart diseases including coronary artery disease, hypertension, cardiomyopathy, arrhythmia, pericardial disease, myocarditis, pulmonary hypertension, myocardial infarction, and many other diseases that can lead to heart failure [103]. Cardiac fibrosis is a hallmark of hypertrophic cardiomyopathy and the basis for HF [29]. In the extracellular space, abnormal accumulation of fibrin and collagen occurs, which often leads to ventricular dysfunction and decreased myocardial stiffness, among others [29]. Cardiac fibroblasts are the main cell type of the heart muscle and are responsible for the synthesis and secretion of collagen in response to the hypertrophic response to pressure overload [104]. Cardiac fibrosis and hypertrophy can make signaling unbalanced leading to the development of the disease, and uncontrolled cardiomyopathy can eventually lead to congestive heart failure (HF) [11,105,106].

Regarding HSPs, a correlation between HF and elevated HSP levels has generally been shown as a result of studies conducted over the past few years. Some recent reports have demonstrated that HSP90 takes part in the regulation of several layers of fibrotic processes through several interacting partners [29]. HSP90 plays a central role in many cardiomyopathy-related pathways. It is involved in cardiac remodeling by inducing hypertrophy and collagen deposition, impairing cardiac function [107,108,109]. A study found that targeting HSP90 has the potential to prevent fibrotic, hypertrophic, and cell death responses, which are key factors in the development of HF [110].

The study found that the inhibition of HSP90 attenuated cardiac hypertrophy in neonatal mouse ventricular cardiomyocytes induced by aortic zonation and phenylephrine, and may be an important target for the treatment of HF [111]. In the heart, HSP90 participates in and regulates numerous pathways that are closely related to HF (Figure 2).

MAPK signaling is associated with the expression of proteins, especially those involved in pathways of cellular inflammation, development, differentiation, proliferation, and apoptosis [112]. MEK5 is involved in the cascade and phosphorylates ERK5, which in turn activates MEF2. Among them, MEF2 is associated with cardiomyocyte hypertrophy [113]. In addition, it was also found that HSP90 can stabilize ERK5 in the cytoplasmic matrix [114]. Major proteins in the PI3K/PKB/mTOR pathway are dependent on HSP90, so this pathway can also be used as a target for regulatory pathways. PI3K signaling is usually initiated by cytokine receptors or by the activation of RTK [115]. Among these proteins, p85, PKB, p110, eIF4E, S6K, and mTOR are all HSP90 clients [116,117,118]. A study found that inhibition of HSP90 can downregulate mTOR and PKB signaling [116,119]. There is also evidence that HSP90 is expressed at higher levels in cardiomyocytes under heat shock through PKB and PKM2 signaling and preserves mitochondrial function via phosphorylation of Bcl2 [120]. Previous studies have shown that activation of calcineurin/calmodulin phosphatase (CaM) dephosphorylates nuclear factor (NFAT) in T cells, which can act as a transcription factor and enter the nucleus. NFAT activation is regulated by GATA and MEF2, both genes associated with cardiac hypertrophy [121]. NFAT has been shown to be associated with cardiac hypertrophy, and, furthermore, its crosstalk with MAPK may enhance pathological results [122,123]. HSP90 stabilizes calmodulin and calcineurin, and when it is inhibited results in attenuated NFAT signaling [124,125]. The internalization of β-adrenergic is closely related to G-protein-coupled receptor kinases (GRKs), β-blockers, and various enzymes involved in endocytic proteins. Moreover, the desensitization of beta-adrenergic receptors also plays a key role in heart disease [126]. HSP90 can bind and stabilize G-protein-coupled receptor kinases GRK6, GRK3, and GRK5, which are all highly expressed in the heart tissues [127,128]. It has been found that when HSP90 is inhibited, GRK expression levels can be reduced by more than 70% [129]. It shows that targeting HSP90 can inhibit the desensitization of β-adrenergic receptors and prevent the development of cardiomyopathy and HF. Tumor necrosis factor alpha (TNFα) signaling activates apoptosis, proliferation, necrosis, and inflammatory responses in cells. This signaling is often associated with myocardial remodeling and is induced in cardiomyocytes by ischemia/reperfusion (IR) injury and HF [110]. TNFα is a common cytokine. It has two receptors, TNFR1 and TNFR2, both of which are expressed in the heart. Both receptors are up-regulated when the heart is damaged by IR [130]. Of these, TNFR1-induced signaling is more associated with apoptosis and necrosis, while TNFR2 responses lead to inflammatory and proliferative responses. TNFR1 has toxic effects on the heart, while TNFR2 plays a maintenance role in cardiac damage [131]. Most of the signaling proteins present in both pathways bind to HSP90. The TNFR1 response is mediated by serine/threonine protein kinase 1 (RIPK1) and TRADD. Two distinct complexes can eventually form, one of which is a necrosome and the other apoptotic. The necrosome formation has been found to be dependent on HSP90, suggesting that HSP90 has a promoting role in necrosis induced by TNFα [132].

Decreased heart function is often associated with the death of some cardiomyocytes, so it is speculated that slowing cardiomyocyte death may prevent the progression of heart failure [133]. Necroptosis is a form of programmed cell necrosis and is associated with cardiomyocyte death during cardiovascular disease [134]. Necroptosis usually occurs in the presence of tumor necrosis factor TNF-α and is tightly and precisely regulated through a series of related signaling pathways [135,136]. Death receptor ligands require interaction with receptor-interacting protein 1 (RIP1) to activate necroptosis [137]. RIP1 normally promotes activation of receptor-interacting protein 3 (RIP3), which then phosphorylate each other. RIP3 acts upstream to phosphorylate RIP1, which in turn mediates downstream RIP3 phosphorylation [138]. Phosphorylated RIP1 and RIP3 induce mixed lineage kinase domain-like (MLKL) activation, and MLKL subsequently translocates to the cytoplasmic membrane [139,140]. Once aggregated in the plasma membrane, MLKL forms permeable pores in the plasma membrane, disrupting its integrity [141,142] and leading to cell necrosis [143]. Multiple studies have shown that MLKL, RIP1, and RIP3 are considered to be the client proteins of HSP90 [144,145,146]. Inhibition of HSP90 may reduce the number of necrotic cells, which usually results in decreased phosphorylation of MLKL [132,147]. In experiments using TAC mice as model animals for HF, it was found that MLKL, RIP1, RIP3, and their phosphorylated forms were elevated after TAC [148]. Caspase-8 is a negative regulator associated with RIP1–RIP3–MLKL signaling. It also has reduced activation at this time [148]. Thus, it can be shown that activation of RIP1–RIP3–MLKL signaling contributes to the process of HF [148]. The increase in MLKL, RIP1, and RIP3 proteins was reduced by using the HSP90 inhibitor 17-AAG, and the phosphorylation of these proteins was also attenuated [148]. Cardiac impairment is also alleviated, as are cardiac hypertrophy and fibrosis [148]. These results reveal that HSP90 can regulate MLKL, RIP1, and RIP3 activity in HF. Consistent with this, recent studies have shown that inhibition of HSP90 can reduce cardiac hypertrophy by inhibiting the RIP1–RIP3–MLKL pathway, thereby improving therapeutic efficacy [149].

## 4. HSP90 Inhibitors Associated with the Treatment of Cardiovascular Disease

Inhibition/regulation of HSP90 has emerged as a novel and potential target for the treatment of cardiovascular disease. In the past, many interventions such as drugs that modulate HSP90 function and expression have emerged to treat cardiac disease (Table 1).

### 4.1. Geldanamycin

Geldanamycin, a natural product, is a first-generation HSP90 inhibitor [66]. Geldanamycin was first identified from Streptomyces in 1970 and was the first benzoquinone ansamycin antibiotic [157]. The application of geldanamycin in IR and ischemic postconditioning (IpostC) rat models show cardioprotective and myocardial anti-inflammatory properties [158]. Inhibition of HSP90 is associated with numerous pathways involved in pathological fibrosis, hypertrophic cells, and death responses in the heart [110]. Under the action of TGF-β, HSP90 induces profibrotic gene expression in the heart. Inhibition of HSP90 using geldanamycin or inhibitory peptides has been found to prevent signaling in profibrotic TGF-β cardiomyocytes and cardiac fibroblasts [109,150]. Activation of NF-κB leads to the expression of proinflammatory cytokines that play a role in cardiac hypertrophy and cardiac fibrosis. Several studies have suggested that geldanamycin treatment disrupts NF-κB signaling by disrupting TNFα signaling [145,151]. Gap junction (GJ) channels provide the basis for intercellular communication in the cardiovascular system to maintain normal heart rhythm, regulate vascular tone and endothelial function, and metabolic exchange between cells. They allow the transfer of small molecules and possibly the diffusion of slow calcium waves; the transfer of “death” or “survival” signals [159]. In cardiomyocytes, the most abundant isoform is connexin 43 (Cx43). It has been shown that ischemic preconditioning induces the translocation of Cx43 from the cytoplasmic matrix to the mitochondria by mechanisms involving HSP90 and outer membrane transposases [152]. Furthermore, in another study, it has been demonstrated that the reduction in mitochondrial Cx43 by geldanamycin is associated with the abrogation of the cardioprotective agent of diazoxide against ischemia-reperfusion-induced cell death [160]. It has also been found that one of the major cellular responses elicited by geldanamycin is the degradation of HSP90 client proteins and the induction of molecular chaperones such as HSP70 [161]. Among them, the abundant HSP70 and HIF-1α and their metabolic target genes are related to cell protection [162]. However, geldanamycin has certain limitations [163]. First, in previous studies in animal models, this drug was found to cause severe hepatotoxicity in therapy due to the metabolism of the benzoquinone moiety even at recommended doses, limiting the effective dose. Therefore, this drug is not currently allowed for clinical treatment [164]. Second, geldanamycin is insoluble in water and is metabolically unstable [164]. Therefore, many variants of geldanamycin have appeared, including 17-AGG, 17-DMAG, radicicol, novobiocin, etc., most notably by changing the quinone ring structure [164]. These developed drug derivatives lead to improved efficacy, tolerability, water solubility, and metabolic stability [165].

### 4.2. 17-AGG

The drug 17-(allylamino)- 17-demethoxygeldanamycin (17-AAG), the first HSP90 inhibitor derived from geldanamycin, was clinically tested in 1999 [66]. Recently, 17-AAG was shown to attenuate atherosclerotic plaque formation by reducing inflammation and inhibiting vascular smooth muscle cell migration and proliferation [102,166]. Studies have shown that 17-AAG can improve pulmonary vascular remodeling by inhibiting the hyperproliferation and migration of PASMCs [74]. Interestingly, HSP90 is also a key regulator in maintaining mitochondrial homeostasis, contributing to the vascular remodeling of PAH, and HSP90 accumulation in PASMC mitochondria is a hallmark of PAH development [88]. Moreover, 17-AAG attenuates hypertrophic remodeling of cardiomyocytes during the development of heart failure. In a rat model, coronary arteries were ligated to induce myocardial infarction in rats and injected 2 to 8 weeks after myocardial infarction. The administration of 17-AAG was found to reduce cardiac insufficiency, hypertrophy, and fibrosis at week 8 after coronary artery injury (CAL), while attenuating the expression of RIP1, RIP3, and MLKL in non-infarcted left ventricular myocardium regions with increased phosphorylation levels. The effect of 17-AAG on cardiac function in rats after myocardial infarction is related to the attenuation of RIP1, RIP3, and MLKL pathways [149]. The CAL-induced cardiac weight and the cross-sectional area also began to diminish after 17-AAG treatment [107]. Rats with CAL showed signs of chronic heart failure and cardiac hypertrophy, and cardiac function was not reduced after treatment with 17-AAG [107]. One study investigated pressure-stressed cardiac hypertrophic mice prepared by constricting the transverse aorta (TAC), which induced significant cardiac fibrosis. After TAC, it was treated with 17-AAG. In these treated animals, the degree of fibrosis was observed after histological staining and their fibrotic volume was found to be reduced [167]. The levels of both dephosphorylated NFATc2 (a transcription factor in cardiac fibroblasts) and calcineurin (HSP90 client protein) were downregulated in contractile transverse aorta (TAC) mice after treatment with 17-AAG [167]. Additionally, ERK and c-Raf signaling, markers of collagen synthesis and cell proliferation, were also attenuated [79]. Therefore, 17-AAG plays an important role in cardiac fibrosis during the development of heart failure [167]. However, 17-AAG has not been approved despite having been tested in clinical trials [168]. The emergence of more potent and water-soluble derivatives such as IPI-504 or 17-DMAG may improve the utility of HSP90 inhibitors as therapeutics.

### 4.3. 17-DMAG

HSP90 inhibitor 17-Dimethylaminoethylamino-17-demethoxygel danamycin (17- DMAG) is also an analog of geldanamycin, which some studies have shown to have anti-inflammatory and anticancer effects [169,170,171]. In addition, there are certain clinical studies on cardiovascular disease. The use of 17-DMAG attenuated ERK activation and suppressed oxidative stress in atherosclerosis. In vitro, VSMCs can be stimulated by 17-DMAG to reverse the phosphorylation of ERK1/2 induced by TNF-α [153]. In addition, decreased levels of ERK1/2 activation in atherosclerotic plaques were also observed through the administration of 17-DMAG in mice models [153]. The use of 17-DMAG reduces ERK phosphorylation in different cell types [172]. NADPH oxidase, a major source of ROS in vascular cells, is also normally induced by TNF-α. One study also found that 17-DMAG also reduced NADPH oxidase activity [154]. Nox1 is the catalytic subunit of NADPH, and HSP90 interacts with it through the C-terminal residue of Nox1. Therefore, the binding of 17-DMAG to HSP90 prevents Nox1 from interacting with HSP90 [154]. These demonstrate that 17-DMAG can reduce oxidative stress in experimental atherosclerosis [153]. It has been found that in aortic tissue, Nrf2 is associated with NF-κB inhibition in atherosclerotic plaques and a reduction in inflammatory components and lesion size, whereas 17-DMAG induces Nrf2 activation. Furthermore, the effects of 17-DMAG on atherosclerosis were also associated with autophagy, antioxidants (catalase, superoxide dismutase, and heme oxygenase-1), and the induction of HSP70 in the aorta of diabetic mice in the organization [53]. In addition, 17-DMAG can also inhibit the formation of cardiac fibrosis; 17-DMAG dephosphorylates dynamin 1-like protein (Drp1) by regulating calcineurin, inhibits AngII-induced mitochondrial fission, and converts exogenous fibroblasts into myofibroblasts [45]. This further supports the tight association of traditional pathways of fibroblast activation, such as AngII and TGFβ, with recently identified mitochondrial and metabolic mechanisms of myocardial fibrosis [46].

### 4.4. Gamitrinib

Gamitrinib selectively targets HSP90 in mitochondria (mtHSP90) and was found to correlate with antiproliferative activity in preclinical models such as fawn-hooded and monocrotaline rats without significant organ or systemic toxicity [173]. The accumulation of mtHSP90 is a hallmark of vascular remodeling in PAH, which may be an exploitable weakness. Inhibition of mtHSP90 with gamitrinib impairs the expression of mitochondria-associated proteins such as mitochondrial large subunit protein (MRPL) and era-like 12S mitochondrial ribosomal RNA chaperone 1 (ERAL1) [87], negatively affects mtDNA-encoded gene expression, and is critical for maintaining OXPHOS capacity [155]. Gamitrinib has been shown to selectively inhibit mtHSP90, which in turn reduces PASMC viability and proliferation. However, it had no effect on control cells, highlighting the pro-activation capacity and disease specificity of mtHSP90 in PASMCs from PAH patients [88]. Inhibition of mtHSP90 by gamitrinib will subsequently reverse pulmonary vascular remodeling and improve cardiac output without apparent toxicity in both PAH models [88]. Therefore, the use of gamitrinib to inhibit HSP90 is an effective way to improve the symptoms of PAH patients and has great advantages in PAH treatment [88].

### 4.5. Celastrol

HSP90 forms homodimers through its CT domain and aggregates with other partners and essential cofactors to form large protein complexes, including p23 and cell division cycle 37 (CDC37). Many client proteins bind to the HSP90 complex, allowing it to be properly folded, activated, transported, stabilized, and even degraded [174]. Celastrol, a medicinal molecule isolated from plant extracts, has shown antioxidant and anti-inflammatory properties in vivo as well as in various cell cultures [175,176,177,178]. Celastrol blocks the binding of HSP90 to CDC37 and is considered an HSP90 inhibitor [179]. To determine celastrol’s effect on the HSP90 pathway, one study tested whether celastrol reduced the protein levels of HSP90 clients. It was found that it decreased the protein levels of FLT3, EGFR, and BCR-ABL1 in several cell lines in a concentration-dependent manner. These findings suggest that celastrol reduces levels of a range of HSP90 client proteins [180]. Given its inhibitory effect on HSP90 clients, it was next investigated whether it affected the activity of HSP90 itself. LNCaP and K562 cells were treated with celastrol and the ATP binding activity of cellular HSP90 was tested. The results show that celastrol directly or indirectly inhibits HSP90 activity in the cellular environment [180]. Furthermore, consistent with HSP90 inhibitory activity, celastrol has also been shown to induce HSP70 levels [181]. Classical inhibitors such as geldanamycin and radicicol have been reported to cause the widespread release of client proteins that undergo transient activation prior to degradation by the ubiquitin–proteasome system (UPS), while others such as celastrol target the co-chaperone complex CDC37 modulators that can then avoid similar large-scale client disruption scenarios by inducing the activation of the client and heat shock response (HSR) [182]. In addition, celastrol reduces inflammation, inhibits NF-κB, and reduces macrophage infiltration in myocardial infarction [156]. In a rat model of permanent coronary occlusion, celastrol inhibits adverse remodeling, reduces infarct size, improves cardiac function, reduces macrophages during chronic preconditioning inflow, and triggers the expression of key cytoprotective proteins including HO-1 [24]. In a rat myocardial model, celastrol can trigger tissue and systemic expression of the cardioprotective protein HO-1. In ischemic myocardium, celastrol improves cardiac function and, in addition to reducing infarct size and protecting cardiac cells from death, celastrol treatment reduces macrophage infiltration, myofibroblast numbers, and reduces fibrosis. Additionally, it inhibits the upregulation of TGF-β and collagen genes observed in adverse cardiac remodeling [24]. In vivo, celastrol treatment for 14 days was well tolerated in rats. Clinical biochemical analysis of plasma samples showed that compared with the saline-treated group the plasma samples had higher levels of uric acid, creatinine, aspartate aminotransferase (AST), alanine aminotransferase (ALT), and γ-glutamyl transfer. Biomarkers such as enzymes, troponin T (TNT), and carrier-processing enzymes (GGT) were not different. In addition, histopathological analysis also showed no tissue damage. Results showed no differences in liver, kidney, and heart health [24]. In rat cardiomyocyte-like H9C2 and primary rat cardiomyocytes, celastrol attenuates Ang II-induced fibrotic and cellular hypertrophic responses. Celastrol directly binds and inhibits STAT3 phosphorylation and nuclear translocation, providing protection from celastrol by targeting STAT3 [183]. A recent study combined in vitro cell culture of a rat cardiomyocyte cell line exposed to ischemia and IR stress and an isolated Langendorf rat heart perfusion IR model. The cardioprotective effects of celastrol were assessed. The results showed that celastrol achieved cardioprotection by activating the cytoprotective HSP and HO-1. It was the most effective in inducing cytoprotective HSP70 and HO-1 protein overexpression and in vitro cell survival. Celastrol is protective against ischemia when administered as pre-treatment or at the time of reperfusion, and it prevents IR injury [180], increases viability, and reduces mitochondrial permeability transition pore openings [184]. In conclusion, celastrol is safe and highly effective, and these results contribute to its development as a novel potent cardioprotective agent and provide new opportunities for clinical development.

### 4.6. Inhibitors of HSP90 and Heart Ischemia

Here we separate HSP90 inhibitors and heart ischemia into separate subsections because we found the data presented to be quite contradictory. In some studies, inhibitors of HSP90 exert cardioprotective effects, while others have found a role in the exacerbation of ischemia.

Myocardial IR injury can be severe or even fatal, significantly reducing the benefit of revascularization in acute myocardial infarction. Several studies have found that inhibitors of HSP90 exert cardioprotective effects in ischemic heart disease. In cardiomyocytes, the activation of TNFR1 is induced by IR injury, and TNFR1-induced signaling is associated with apoptosis and necrosis with toxic effects on the heart [111,132]. The necrosome formation has been found to be dependent on HSP90, suggesting a promoting role for HSP90 in TNFα-induced necrosis. The application of geldanamycin in IR and IpostC rat models showed cardioprotective and myocardial anti-inflammatory properties [159]. Several studies have also shown that geldanamycin treatment disrupts NF-κB signaling by disrupting TNFα signaling [146,152]. A recent study evaluating the cardioprotective effects of celastrol found that celastrol, when administered as preconditioning or reperfusion, prevented ischemia and prevented IR injury [181].

In addition, some studies have also found that HSP90 has a profound effect on IpostC cardioprotection, which may be related to the inhibition of the complement system and JNK, ultimately attenuating I/R-induced myocardial injury and apoptosis. Conversely, geldanamycin treatment reversed the protection of IPC [185]. HSP90 is critical for morphine postconditioning (MP)-mediated cardioprotection, possibly by promoting phosphorylation of Akt, inhibiting activation of complement component 5a (C5a) and NF-κB signaling, and subsequent myocardial inflammation, ultimately attenuating infarct size and cardiomyocyte apoptosis. However, the HSP90 inhibitor geldanamycin or the Akt inhibitor GSK increased the expression of C5a and NF-κB and prevented MP-induced cardioprotection [186].

Because of the conflicting results of HSP90 inhibitors on ischemic heart disease, researchers need to be aware that the unclear effects of HSP90 inhibitors on ischemic heart disease should be considered in the context of HSP90 inhibitor-based drugs for the treatment of heart disease.

## 5. Conclusions

According to the World Health Organization, cardiovascular disease ranks first in the world in terms of morbidity and mortality, and about 23.6 million people will die from these diseases by 2030 [187]. HSP90 is closely related to many cardiovascular diseases. Numerous client proteins of HSP90 have been identified in known cardiac disease pathways, including MAPK signaling, PI3K/AKT (PKB)/mTOR, and TNF-α signaling, all of which are direct and indirect targets of HSP90. HSP90 participates extensively in these pathways through its activity and stability. Understanding the proteins that interact with HSP90 can help reveal its possible role in these pathways.

While the functions of inhibiting HSP90 have not been studied in all settings, they have been investigated in many studies and the effects of inhibition on certain pathways can be demonstrated. Promising results from many studies and demonstrated efficiencies in different types of cell cultures and mouse models suggest that HSP90-based therapies have great potential. We could provide a novel therapeutic approach by modulating HSP90 activity rather than systemic inhibition. However, HSP90 inhibitors are toxic, and currently HSP90 inhibitors have not been allowed on the market and are still in the clinical/preclinical experimental stage [66,169,170,171,173,174]. They can also lead to the development of unknown side effects when used in combination with other drugs. In addition, the unclear effects of HSP90 inhibitors on the ischemic heart should also be considered in the context of HSP90 inhibitor-based drug therapy for cardiac disease. Much remains to be learned about their molecular mechanisms of action, and the future development of safe, healthy, and effective HSP90 inhibitors is challenging. Overall, the novel class of HSP90 inhibitors holds great potential for the treatment of cardiovascular disease. This area of expertise specifically targeting HSP90 in cardiovascular disease appears to be still fresh and yet to be further investigated. Therefore, we believe that this area will generate further interest.

## Figures and Tables

**Figure 1 cells-11-03444-f001:**
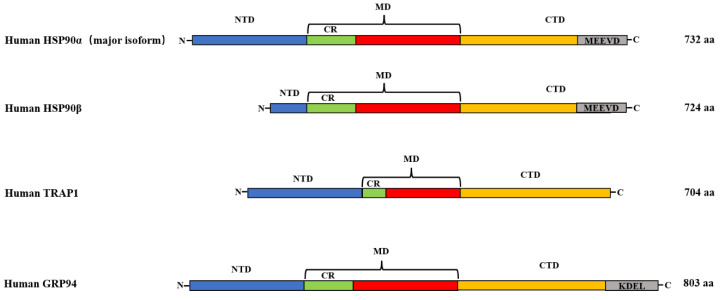
Diversity of protein domains of human HSP90 family members. HSP90α and HSP90β are mainly in the cytoplasm, TRAP1 is mainly in the mitochondrial matrix, and GRP94 is mainly in the ER; HSP90α, HSP90β, GRP94, and TRAP1 are, respectively, 732, 724, 803, and 704 amino acids in length. Their overall molecular structure includes three major conserved domains: NTD (ATPase, co-chaperone binding), MD (client protein binding, co-chaperone binding, ATPase activation), and CTD (dimerization, co-chaperone binding, client protein binding). CR in the above figure represents charged flexible linker.

**Figure 2 cells-11-03444-f002:**
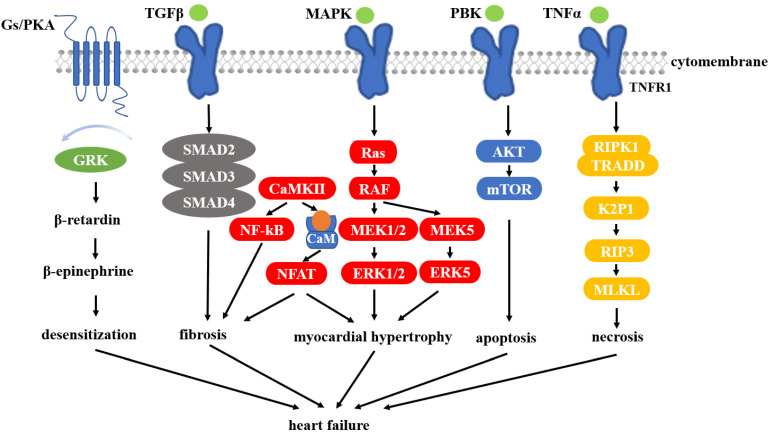
Pathways associated with heart failure.

**Table 1 cells-11-03444-t001:** HSP90 inhibitors associated with the treatment of cardiovascular disease.

Drug	Disease/Model	Detection Indicator	References
Geldanamycin	Ventricular hypertrophy; myocardial fibrosis; heart failure	TGF-β; NF-κB; Cx43	[150,151,152]
17-AGG	Atherosclerosis; pulmonary hypertension; heart failure	RIP1; RIP3; MLKL; c-Raf; ERK	[142,149]
17-DMAG	Atherosclerosis; myocardial fibrosis	ERK; NADPH oxidase; NF-κB; Drp1	[108,117,153,154]
Gamitrinib	Pulmonary hypertension	MRPL; ERAL1	[87,155]
Celastrol	Myocardial infarction; ischemic cardiomyopathy	NF-κB; HO-1	[156]

## Data Availability

Not applicable.

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
