# Peer review of "The Role of HSP90 Inhibitors in the Treatment of Cardiovascular Diseases"

_cells, 2022, doi:10.3390/cells11213444_

Round 1
Reviewer 1 Report
The manuscript of Shiyu Qi et al. is dedicated to a literature review on recent data on involvement of heat shock protein 90 kDa (HSP90) in cardiovascular diseases and some possibilities of inhibitor-induced modulation of HSP90 expression for treatment of heart disease. This field of investigation is significant because cardiovascular diseases are surely an important biomedical problem and clarification of poorly understood mechanisms of the diseases is of importance for development of novel effective therapeutic approaches to treat these diseases. The significance of this review is considerable, since the presented summary of literature data and their analysis give an objective integral picture of existing ideas on the indicated topic and highlight possible ways of practical use of HSP90 inhibitors in cardiology. These include information concerning the role of HSP90 in the main varieties of cardiovascular diseases, and description of HSP90 inhibitors associated with the treatment of the diseases.
In general the review is well presented. However, the manuscript contains a number of inaccuracies and misprints in the text. The authors should correct the mistakes. I am also surprised that the authors of the review do not have publications on the topic of this review.
Besides, I have some minor comments that might improve the overall quality of the manuscript:
1. Page 2, line 55. “…cardiovascular disease …” Here the abbreviation (CVD) should be inserted.
2. Page 2, line 60-61. “ Mortality from cardiovascular disease is largely determined by disease caused by atherosclerosis. Among them, atherosclerosis is the main cause of death from cardiovascular disease. “ These two sentences bring the same information, so authors should delete the second one.
3. Page 3, line 110. “ Both HSP90α and HSP90β are found in the cytoplasmic matrix. “ The previous sentence contains already this information, so authors should delete this sentence.
4. Page 6, line 239-240. “ In addition, the study also found that the beneficial effect of inhibiting HSP90 activation was related to the overexpression of HSP70. “ It is very important information for the theme of the review, because this compensatory reaction of cells to treatment with HSP90 inhibitors is the real barrier for effective application of these preparations in biomedicine. So, it would be reasonable to discuss this fact in “Conclusion”.
5. Page 7, line 297-298. “ Signaling Tumor necrosis factor alpha (TNFα) signaling “ This is just one example of the inadvertence of the authors.
Author Response
Response to reviewer1:
We would like to thank you for your careful reading, helpful comments and constructive suggestions, which have greatly improved the presentation of our manuscript. We have carefully considered all the comments of the reviewers and have revised our manuscript accordingly. First, we have thoroughly checked and revised our manuscript for errors in grammar, fonts, etc. In addition, we have done some studies related to HSP90 before, and some related studies on HSP90 and cardiovascular disease are ongoing. Finally, in the next section, we summarize our responses to each of the reviewers' comments. We believe our response addresses all of the reviewer's concerns well. We hope that our revised manuscript will be accepted for publication.
- Page 2, line 55. “…cardiovascular disease …” Here the abbreviation (CVD) should be inserted.
Response: This content has been changed, see line 56
- Page 2, line 60-61. “ Mortality from cardiovascular disease is largely determined by disease caused by atherosclerosis. Among them, atherosclerosis is the main cause of death from cardiovascular disease. “ These two sentences bring the same information, so authors should delete the second one.
Response:The second has been removed.
- Page 3, line 110. “ Both HSP90α and HSP90β are found in the cytoplasmic matrix. “ The previous sentence contains already this information, so authors should delete this sentence.
Response: This content has been removed
- Page 6, line 239-240. “ In addition, the study also found that the beneficial effect of inhibiting HSP90 activation was related to the overexpression of HSP70. “ It is very important information for the theme of the review, because this compensatory reaction of cells to treatment with HSP90 inhibitors is the real barrier for effective application of these preparations in biomedicine. So, it would be reasonable to discuss this fact in “Conclusion”.
Response: This content has been changed.
- Page 7, line 297-298. “ Signaling Tumor necrosis factor alpha (TNFα) signaling “ This is just one example of the inadvertence of the authors.
Response: This content has been changed.
Reviewer 2 Report
In the present review entitled “The role of Hsp90 inhibitors in the treatment of cardiovascular diseases” by Qi et al., the authors highlight the role of HSP90 in the treatment of cardiovascular diseases, primary causes of death in the world.
Even if the subject of the paper is interesting and actual, there are too many editing errors, and above all the text requires extensive revision from a native speaker.
For all these reasons the paper cannot be accepted in the present form.

Author Response
Response to reviewer2:
We really appreciate your reading, there are so many details we didn't notice before. Your helpful comments to us will greatly enhance the presentation of our manuscript. We have carefully considered all the comments of the reviewers and made comprehensive revisions in some aspects of language, grammar, format, etc. In the next section, we summarize our responses to each of the reviewers' comments. We hope that our revised manuscript will be accepted for publication.
- Line 23: “Finally, we summarize” → Finally, we will summarize
Response: This content has been changed, see line 24.
- Lines 35-36: “HSP expression and synthesis are inhibited following stress” → HSP expression and synthesis are increased following stresses
Response: This content has been changed, see line 36,37.
- Line 41: “As a molecular chaperone, its synthesis” → As molecular chaperones, their synthesis
Response: This content has been changed, see line 42.
- Line 43: “three-dimensional structure” → three-dimensional protein structure
Response: This content has been changed, see line 44.
- Line 44: “achieve normal physiology function” → achieve normal physiological function
Response: This content has been changed, see line 45.
- Lines 44-45 : “HSPs are also involved in 44 controlling the transport of certain regulatory proteins and intracellular proteins”. What does it mean? Please better clarify
Response: This content has been changed, see line 46.
- Line 66: “function is to continuously supply nutrient tissues and blood to other tissues” → function is to continuously supply nutrients and blood to other tissues
Response: This content has been changed, see line 68
- Line 67: “maintain its physiological metabolism and play a central role in the normal physiology”
Response: This content has been changed, see line 67
- Line 90: “reducing its aggregation with other” → reducing their aggregation with other
Response: This content has been changed, see line 92
- Lines 96-97: “N-terminal domain are called nucleotide binding sites” → N-terminal domain is called nucleotide binding site
Response: This content has been changed, see line 99
- Lines 97-98: “The C-terminal domain contains MEEVD or KDEL, a special protein responsible for protein dimerization”
Response: This content has been changed, see line 100
- Lines 103-104: “HSP90α, HSP90β, GRP94 and TRAP1 are 732, 724, 803 and 704 in length, respectively amino acids; its overall molecular structure”
Response: This content has been changed, see line 104
- Line 109: “may be caused by”
Response: This content has been changed, see line 110
- Lines 111-116 rewrite better
Response: This content has been changed, see line 115-116
- Line 117-118: “most client proteins are able to bind to either.” → most client proteins are able to bind to both of them.
Response: This content has been changed, see line 117
- Lines 120-122: “GRP94 is another isoform of HSP90 and shares 50% homology with cytoplasmic HSP90[32]. Most abundant in the endoplasmic reticulum, also known as endoplasmin[35,36].”
Response: This content has been changed, see line 119-121
- Line 127: “but its length is shorter than that of HSP90 in cells” → but its length is shorter than that of HSP90 in cytoplasm
Response: This content has been changed, see line 126
- Lines 142-143: “On the other hand, compared with the protein HSP90, TRAP1 lacks” → On the other hand, compared to the protein HSP90, TRAP1 lacks
Response: This content has been changed, see line 142
- Lines 153-155: “One study evaluated serum HSP90α concentrations in patients with arterial hypertension versus normotensives, and serum HSP90α concentrations in patients with arterial hypertension were significantly higher than their normotensive counterparts [53].”
Response: This content has been removed
- Line 157: “impaired nitric oxide bioavailability [53]. Confirmed that” → impaired nitric oxide bioavailability [53], confirming that
Response: This content has been changed, see line 162
- Lines 172-173: “AngII-induced phenotype switching in atrial fibrillation and subsequent migration and proliferation [58].” → please better clarify the subject of migration and proliferation
Response: This content has been changed, see line 176-178
- Lines 189-190: “A Study has shown that the levels of HSP90 are increased in both the plasma membrane wall and the membrane wall of pulmonary arterioles in PAH patients [63].” → Please clarify the meaning of plasma membrane wall and membrane wall of pulmonary arterioles
Response: This content has been changed, see line 196-197
- Lines 214-215: “It is considered an autoimmune chronic disease that usually culminates due to the formation of plaques [81].” rewrite better
- Response: This content has been changed, see line 224
- Line 225: “play a key role in the formation of athero-sclerosis” → atherosclerosis does not “form” so maybe you can modify:
- Response: This content has been changed, see line 235
- Line 229: “HSP90 was found to be improved in areas of inflammation”. What does improve means? Please clarify
Response: This content has been changed, see line 239-240
- Line 233: “mice treated with HSP90 inhibitors” → treatments with HSP90 inhibitors
Response: This content has been changed, see line 243
- Lines 252-254: “Cardiac fibroblasts are the main cell type of the heart muscle and are responsible for the synthesis and secretion of collagen in response to the hypertrophic response to pressure overload [91].” rewrite better
Response: This content has been changed, see line 263-265
- Lines 258-260: “Several recent reports have demonstrated that HSP90 take part in the regulation of several layers of fibrotic processes and through several interacting partners [21].” → the word “several” is used too many times, please modify
Response: This content has been changed, see line 269
- Line 273: “MEK5 involved in the cascade” → MEK5 is involved in the cascade and
Response: This content has been changed, see line 284
- Line 309: “suggest that HSP90 has a promoting role in necrosis induced by TNFα [119].” → suggesting that HSP90 has a promoting role in necrosis induced by TNFα [119]
Response: This content has been changed, see line 320
- Lines 317-319: “RIP1 normally promotes activation of receptor interacting protein 3 (RIP3), which then phosphorylate each other [125].” It is not clear who phosphorylates who. Please clarify
Response: This content has been changed, see line 328-335
- Line 346: “IPostC” what’s the meaning of this word?
Response: IPostC is short for ischemic postconditioning ,This content has been changed, see line 359
- Lines 349-350: “HSP90 induces profibrotic gene expression in 349 the heart in the context of TGF-β.” The meaning of this phrase is not clear, please clarify
Response: This content has been changed, see line 361-364
- Lines 375-376: “17-(allylamino)-17-demethoxygeldanamycin (17-AAG), the first Hsp90 inhibitor, was clinically tested in 1999, derived from geldanamycin[60].” → 17-(allylamino)- 17-demethoxygeldanamycin (17-AAG), the first Hsp90 inhibitor derived from geldanamycin, was clinically tested in 1999 [60].
Response: This content has been changed, see line 386
- Lines 416-419 rewrite better
Response: This content has been changed, see line 428-431
- Lines 423-425 rewrite better
Response: This content has been changed, see line 432-437
- Line 433: “preclinical model”. Please specify which models they are
Response: This content has been changed, see line 445-446
- Lines 469-470: “In vivo, celastrol treatment was well tolerated in rats for 14 days.” → In vivo, celastrol treatment for 14 days was well tolerated in rats.
Response: This content has been changed, see line 491-492
- Lines 470-475 The meaning is not clear, please clarify
Response: This content has been changed, see line 471-476
- Line 492: “Promising results from many studies and demonstrated” → it seems that something is missing.
Response: This content has been changed, see line 523
- Line 497: “HSP90 inhibitors have not been allowed to market and are still in the clinical/preclinical experimental stage” → please add references about HSP90 inhibitor clinical trials
Response: References to clinical trials of HSP90 inhibitors have been added.
Minor revisions:
- Based on your comments, We have conducted a comprehensive review of the article for issues such as spacing, punctuation, grammar and spelling errors. and added the meaning of abbreviations.
- We have made revisions to ill-expressed sentences in the manuscript.
- For the convenience of readers, the language of the entire article has been improved.
Reviewer 3 Report
Deng et al in their review present data on HSP90 function in cardiac diseases and potential drugs able to modulate the chaperone function in above pathologies. Analyzing the expertise of the authors one may see that two corresponding persons published a couple of papers on HSP90 function in ovarian theca cells or in human semen, while all four authors recently published review on drugs targeting tumor angiogenesis. So, I am not sure that the transition to a new topic is good enough for publication in well-established journal, such as Cells.
The m/s is full of wrong statements, grammatical errors and flaws. I started from the beginning and my feeling is that the m/s should be corrected as the whole. Below, I present a few of points that explain my position to the m/s, its real state.
Major concerns
--Line 154: “serum HSP90α concentrations in patients with arterial hypertension were significantly higher than their normotensive counterparts [53]” – It is fully unclear why the mainly cytoplasmic protein HSP90 was found in serum? Did it leak from dying cells in a manner of LDH or cardiac troponin and being a sign of cell death or its appearance in extracellular matrix has another explanation. A few of studies show that a part of HSP90 molecules can occur outside living cells.
--Inhibitors of HSP90
Long story of HSP90 inhibitors tells us that they are mostly employed in cancer treatment, and even in this field they meet mixed opinion since many of those are able to give the effect of HSP70 induction which is evidently powerful cytoprotective effect; hundreds of studies are dedicated to this phenomenon.
--Animal models of HSP90 and/or cardiac pathology
Scanning the text, I found almost no data (maybe one or two) obtained with the use of above models; it’s a sad omission.
--Chapter on celastrol
According to well established data celastrol regulates HSF1 activity and therefore enhances the expression of various chaperones including HSP90; it is shown that the major therapeutic target for celastrol is HSP70 which has a great cytoprotective potential proved in large number of studies on myocardial pathologies. So, instead the very particular cases of celastrol effects on HSP90-based machinery, the authors should present more prevalent data concerning the therapeutic effects of the drug
I can present even more claims to the authors, however, I hope that they are able to make conclusions themselves.
Minor comments
Line 35 – It is difficult to understand what is hidden behind the following phrase: Unlike other mostly cellular proteins, HSP expression and synthesis are inhibited following stress such as oxidative stress
Line 73 - What are “immune-avidin co-chaperones”?
Lines 74 – “HSP90…can also destroy and degrade mis-folded proteins [2,24]”. Does HSP90 really destroy such proteins?
Line 75 – “Therefore, HSP90 plays an important role in the treatment and research of heart and vascular diseases.” – How can a protein “play an important role” in treatment and research?
Line 97 – “The C-terminal domain contains MEEVD or KDEL, a special protein responsible…” – DOMAIN or PROTEIN?
Line 115: “Although HSP90α and HSP90…”??
Line 166: what is CN?
Line 173: A study compared mice with different diet-induced hypertension regimens, with the addition of repetitive hyperthermia in some groups [59] – What is result of the study?
Line 229: HSP90 was found to be improved in areas of inflammation – improved??
At this point I was forced to stop studying the text, because I don’t want to take on the work of the authors.
Finally, I would advise the authors to study two recent reviews on HSP90 function in pathologies of cardiac diseases:
-Roberts et al. The Potential of Hsp90 in Targeting Pathological Pathways in Cardiac Diseases. J Pers Med. 2021 Dec 16;11(12):1373. doi: 10.3390/jpm11121373.
-Chakafana et al. Heat Shock Proteins: Potential Modulators and Candidate Biomarkers of Peripartum Cardiomyopathy. Front Cardiovasc Med. 2021 Jun 16;8:633013. doi: 10.3389/fcvm.2021.633013.
Author Response
Response to reviewer3:
We greatly appreciate your careful reading and constructive suggestions that have greatly improved the presentation of our manuscript. We have carefully considered all the comments of the reviewers and have revised our manuscript accordingly.
As you can see we have research on HSP90 and its function in ovarian theca cells or in human semen, and we are currently doing research on HSP90 and cardiovascular disease and so on. In addition, we have comprehensively revised the manuscript for misstatements, grammatical errors, and flaws.
In the next section, we summarize our responses to each reviewer's comments. We believe our response addresses all of the reviewer's concerns well. We hope that our revised manuscript will be accepted for publication. Finally, we are very grateful to the reviewers for providing us with two references that were very helpful to our review. We have carefully studied and applied them to our review.
Major concerns:
- In response to the reviewer's question, we have supplemented the content from 152 lines to 160 lines. The reviewer did not explain why the predominantly cytoplasmic protein HSP90 was found in serum, here we explain that arterial hypertension is associated with impaired NO function. One of the key client proteins of Hsp-90 is eNOS. In spontaneous hypertension, Hsp-90 expression and all isoforms of NOS, including eNOS, were significantly increased. Increased Hsp-90α expression compensates for impaired endothelial NOS function associated with reduced nitric oxide bioavailability.
- According to the reviewer's problem of less use of animal models, we have supplemented, here we point out the animal models we mentioned:3.1. HSP90 and hypertension:Spontaneously Hypertensive Rat Model,3.2 HSP90 and PAH:monoclostaline-induced rat model ,3.3 HSP90 and atherosclerosis :Hyperlipidemic apoE-deficient mice ,3.4. HSP90 and heart failure:TAC mouse model,4.1. Geldanamycin:IpostC rat models,4.2 17-AGG:Rat model and TAC mouse model, etc. 4.3 17-DMAG:Diabetic mouse model,4.4 Gamitrinib:fawn-hooded and monocrotaline rats, 4.5. Celastrol: rat myocardial model.
- As suggested by the reviewers, we have supplemented with more general data on the effects of celastrol treatment, see lines 467 to 476 and 501 to 509.
- Regarding the HSP90 inhibitors mentioned by the reviewer, we are also conducting related research.
- Finally, we are very grateful to the reviewers for providing us with two references that were very helpful to our review. We have carefully studied and applied them to our review.
Minor comments:
- Line 35 – It is difficult to understand what is hidden behind the following phrase: Unlike other mostly cellular proteins, HSP expression and synthesis are inhibited following stress such as oxidative stress
Response: This content has been changed, see line 36-38.
- Line 73 - What are “immune-avidin co-chaperones”?
Response: This content has been changed, see line 72-76.
- Lines 74 – “HSP90…can also destroy and degrade mis-folded proteins [2,24]”. Does HSP90 really destroy such proteins?
Response: This content has been changed, see line 73-76.
- Line 75 – “Therefore, HSP90 plays an important role in the treatment and research of heart and vascular diseases.” – How can a protein “play an important role” in treatment and research?
Response: This content has been changed, see line 73-76.
- Line 97 – “The C-terminal domain contains MEEVD or KDEL, a special protein responsible…” – DOMAIN or PROTEIN?
Response: MEEVD or KDEL are special motifs,This content has been changed, see line 100.
- Line 115: “Although HSP90α and HSP90…”??
Response: This content has been changed, see line 113.
- Line 166: what is CN?
Response: CN is calcineurin, this content has been changed, see line 171.
- Line 173: A study compared mice with different diet-induced hypertension regimens, with the addition of repetitive hyperthermia in some groups [59] – What is result of the study?
Response: The results of the study found that mice treated with a high-salt diet developed cardiac hypertrophy and fibrosis, whereas repetitive hyperthermia attenuated salt-induced cardiac hypertrophy, myocardial and perivascular fibrosis, and blood pressure elevation, see line 180.
- Line 229: HSP90 was found to be improved in areas of inflammation – improved??
Response: This content has been changed, see line239-240.
Round 2
Reviewer 3 Report
On my mind Deng et al have revised their manuscript to the extent how they understand this work, although I asked them to rewrite the article fully. They did not answer my question concerning extracellular HSP90, though there are many reports indicating important role of such form of the chaperone and what is more important did not describe the specificity of the effects of HSP90 inhibitors on HSP70 protective power. This power may be a key element to cardioprotective effects of some of above inhibitors.
Responding to my minor comments the authors monotonously answered "this content has been changed"' and this response they had to repeat tens times, since I stopped to request them to do that; I was really exhausted.
Author Response
We are grateful to the reviewers for re-reviewing our manuscript. We are very sorry that the previous answer did not satisfy the reviewer. We have revised the manuscript again according to the reviewer's suggestion, and hope this time we can answer the reviewer's question well. At the same time, we very much hope that our manuscript will be accepted.
- Explain how the largely intracellular protein Hsp90 can come to blood (serum)?
R:Regarding this issue, we are based on the literature Skorzynska-Dziduszko, K.E.; Olszewska, A.; Prendecka, M.; Malecka-Massalska, T. Serum Heat Shock Protein 90 Alpha: A New Marker of Hypertension-Induced Endothelial Injury? Adv Clin Exp Med 2016, 25, 255-261, doi:10.17219/acem/40068. to describe, this article is indeed based on the comparison of HSP90 in serum. But we really can't give a reasonable explanation about how the intracellular HSP90 gets into the serum. We think that the explanation proposed by the reviewer may be correct, that HSP90 may be leaked from dying cells, because according to the tumor-related literature we reviewed, HSP90α can be specifically targeted by tumors when there is tumor development. Cells are secreted into the blood outside the cell, and its concentration in plasma is closely related to the occurrence and progression of malignant tumors. However, we really have no way of giving references to explain exactly why high blood pressure leads to the increase of HSP90 in serum. We have made reasonable changes to this part of the content, please refer to Line 154-172 of the original text for details. And in the process of our review of the literature, we also found that in many diseases, HSP90 in serum is considered as a prospective biomarker eg: Serum HSP90-Alpha and Oral Squamous Cell Carcinoma: A Prospective Biomarker, etc.
original content: Hypertension-induced endothelial dysfunction is associated with impaired nitric oxide (NO) bioavailability regulated through the interaction between nitric oxide syn-thase and HSP [61]. It is well known that nitric oxide is one of the key factors in main-taining normal blood pressure. Arterial hypertension is associated with impaired NO function [62,63]. One of the key client proteins of HSP90 is endothelial NOS. HSP90 binds to eNOS, thereby determining its correct folding and function [64]. In an animal model of spontaneous hypertension, HSP90 expression was higher in the left ventricle of tonic rats compared with normotensive control animals. Interestingly, all isoforms of NOS (neuronal (nNOS), inducible (iNOS) and eNOS) were initially down-regulated, but were eventually significantly increased in the left ventricle of spontaneously hy-pertensive rats. Increased expression of HSP90α compensates for impaired eNOS function associated with reduced nitric oxide bioavailability [61]. Increased HSP90α concentrations in patients with arterial hypertension may be a compensatory mecha-nism for impaired nitric oxide bioavailability [61], confirming that HSP90α can serve as an early marker of hypertension-related endothelial injury [61]. In addition, it was also found that this study mainly focused on HSP90 in serum. Ac-cording to previous studies in tumors, it was found that when a tumor occurs, HSP90α can be specifically secreted by tumor cells into the blood outside the cell, and its con-centration in plasma is related to malignant tumors. The occurrence and progress are closely related.
2) Discuss that inhibitors Hsp90 can activate HSF1 and thus induce cytoprotective Hsps (including Hsp70, Hsp60, Hsp27 and others) which can exert the cardioprotective effects.
After such additions, your manuscript may be accepted.
R:It is known that Hsp90 has many client proteins that contribute to a group called. Among them are HSF1 and Hsp70 (Line 55). As stated by the reviewers, Hsp90 inhibitors can activate HSF1, thereby inducing cytoprotective Hsps (including Hsp70, Hsp60, Hsp27, etc.) to exert cardioprotective effects. This part is also mentioned several times in our article. In the section on HSP90 and atherosclerosis, we mentioned that the beneficial effect of inhibiting HSP90 activation is related to the overexpression of HSP70, which protects against atherosclerosis. (Line 258). In addition, among HSP90-related inhibitors, we also mentioned that Geldanamycin can induce the level of HSP70, and in addition, HSP70 was found to be associated with cardiac cytoprotection (Line 387-389). The effect of 17-DMAG on atherosclerosis is associated with the induction of HSP70 in the aorta (Line 449). Similarly, Celastrol, an HSP90 inhibitor, has also been shown to induce HSP70 levels (Line 488). It was most effective in inducing cytoprotective HSP70 and HO-1 protein overexpression and in vitro cell survival (Line 517). In addition, we also found that the levels of HSP60, HSP70 and HSP90 were all increased in mice subjected to repeated hyperthermia (Lin 191).